# Semidefinite relaxations
# for certifying robustness to adversarial examples

**Aditi Raghunathan, Jacob Steinhardt and Percy Liang**
Stanford University
{aditir, jsteinhardt, pliang}@cs.stanford.edu

## Abstract

Despite their impressive performance on diverse tasks, neural networks fail catastrophically in the presence of adversarial inputs—imperceptibly but adversarially perturbed versions of natural inputs. We have witnessed an arms race between defenders who attempt to train robust networks and attackers who try to construct adversarial examples. One promise of ending the arms race is developing certified defenses, ones which are provably robust against all attackers in some family. These certified defenses are based on convex relaxations which construct an upper bound on the worst case loss over all attackers in the family. Previous relaxations are loose on networks that are not trained against the respective relaxation. In this paper, we propose a new semidefinite relaxation for certifying robustness that applies to arbitrary ReLU networks. We show that our proposed relaxation is tighter than previous relaxations and produces meaningful robustness guarantees on three different *foreign networks* whose training objectives are agnostic to our proposed relaxation.

## 1 Introduction

Many state-of-the-art classifiers have been shown to fail catastrophically in the presence of small imperceptible but adversarial perturbations. Since the discovery of such adversarial examples [25], numerous defenses have been proposed in attempt to build classifiers that are robust to adversarial examples. However, defenses are routinely broken by new attackers who adapt to the proposed defense, leading to an arms race. For example, distillation was proposed [22] but shown to be ineffective [5]. A proposed defense based on transformations of test inputs [20] was broken in only five days [2]. Recently, seven defenses published at ICLR 2018 fell to the attacks of Athalye et al. [3].

A recent body of work aims to break this arms race by training classifiers that are *certifiably robust* to all attacks within a fixed attack model [13, 23, 29, 8]. These approaches construct a convex relaxation for computing an upper bound on the worst-case loss over all valid attacks—this upper bound serves as a *certificate* of robustness. In this work, we propose a new convex relaxation based on semidefinite programming (SDP) that is significantly tighter than previous relaxations based on linear programming (LP) [29, 8, 9] and handles arbitrary number of layers (unlike the formulation in [23], which was restricted to two). We summarize the properties of our relaxation as follows:

    1. Our new SDP relaxation reasons *jointly* about intermediate activations and captures interactions that the LP relaxation cannot. Theoretically, we prove that there is a square root dimension gap between the LP relaxation and our proposed SDP relaxation for neural networks with random weights.

    2. Empirically, the tightness of our proposed relaxation allows us to obtain tight certificates for *foreign networks*—networks that were not specifically trained towards the certification procedure. For instance, adversarial training against the Projected Gradient Descent (PGD) attack [21] has led to networks that are "empirically" robust against known attacks, but which have only been certified against small perturbations (e.g. $\epsilon = 0.05$ in the $\ell_\infty$-norm for the MNIST dataset [9]). We use our SDP

to provide the first non-trivial certificate of robustness for a moderate-size adversarially-trained model on MNIST at $\epsilon = 0.1$.

3. Furthermore, training a network to minimize the optimum of particular relaxation produces networks for which the respective relaxation provides good robustness certificates [23]. Notably and surprisingly, on such networks, our relaxation provides tighter certificates than even the relaxation that was optimized for during training.

**Related work.** Certification methods which evaluate the performance of a given network against all possible attacks roughly fall into two categories. The first category leverages convex optimization and our work adds to this family. Convex relaxations are useful for various reasons. Wong and Kolter [29], Raghunathan et al. [23] exploited the theory of duality to *train* certifiably robust networks on MNIST. In recent work, Dvijotham et al. [8], Wong et al. [30] extended this approach to train bigger networks with improved certified error and on larger datasets. Solving a convex relaxation for certification typically involves standard techniques from convex optimization. This enables scalable certification by providing valid upper bounds at every step in the optimization [9].

The second category draws techniques from formal verification such as SMT [16, 17, 7, 14], which aim to provide tight certificates for *any* network using discrete optimization. These techniques, while providing tight certificates on arbitrary networks, are often very slow and worst-case exponential in network size. In prior work, certification would take up to several hours or longer for a single example even for a small network with around 100 hidden units [7, 16]. However, in concurrent work, Tjeng and Tedrake [26] impressively scaled up exact verification through careful preprocessing and efficient pruning that dramatically reduces the search space. In particular, they concurrently obtain non-trivial certificates of robustness on a moderately-sized network trained using the adversarial training objective of [21] on MNIST at perturbation level $\epsilon = 0.1$.

## 2 Setup

Our main contribution is a semidefinite relaxation of an optimization objective that arises in certification of neural networks against adversarial examples. In this section, we set up relevant notation and present the optimization objective that will be the focus of the rest of the paper.

**Notation.** For a vector $z \in \mathbb{R}^n$, we use $z_i$ to denote the $i^{\text{th}}$ coordinate of $z$. For a matrix $Z \in \mathbb{R}^{m \times n}$, $Z_i \in \mathbb{R}^n$ denotes the $i^{\text{th}}$ row. For any function $f : \mathbb{R} \to \mathbb{R}$ and a vector $z \in \mathbb{R}^n$, $f(z)$ is a vector in $\mathbb{R}^n$ with $(f(z))_i = f(z_i)$, e.g., $z^2 \in \mathbb{R}^n$ represents the function that squares each component. For $z, y \in \mathbb{R}^n$, $z \succeq y$ denotes that $z_i \geq y_i$ for $i = 1, 2, ..., n$. We use $z_1 \odot z_2$ to represent the elementwise product of the vectors $z_1$ and $z_2$. We use $B_\epsilon(\bar{x}) \overset{\text{def}}{=} \{x \mid \|x - \bar{x}\|_\infty \leq \epsilon\}$ to denote the $\ell_\infty$ ball around $\bar{x}$. When it is necessary to distinguish vectors from scalars (in Section 4.1), we use $\vec{x}$ to represent a vector in $\mathbb{R}^n$ that is semantically associated with the scalar $x$. Finally, we denote the vector of all zeros by $\mathbf{0}$ and the vector of all ones by $\mathbf{1}$.

**Multi-layer ReLU networks for classification.** We focus on multi-layer neural networks with ReLU activations. A network $f$ with $L$ hidden layers is defined as follows: let $x^0 \in \mathbb{R}^d$ denote the input and $x^1, ..., x^L$ denote the activation vectors at the intermediate layers. Suppose the network has $m_i$ units in layer $i$. $x^i$ is related to $x^{i-1}$ as $x^i = \text{ReLU}(W^{i-1} x^{i-1}) = \max(W^{i-1} x^{i-1}, 0)$, where $W^{i-1} \in \mathbb{R}^{m_i \times m_{i-1}}$ are the weights of the network. For simplicity of exposition, we omit the bias terms associated with the activations (but consider them in the experiments). We are interested in neural networks for classification where we classify an input into one of $k$ classes. The output of the network is $f(x^0) \in \mathbb{R}^k$ such that $f(x^0)_j = c_j^\top x^L$ represents the score of class $j$. The class label $y$ assigned to the input $x^0$ is the class with the highest score: $y = \text{argmax}_{j=1,...,k} f(x^0)_j$.

**Attack model and certificate of robustness.** We study classification in the presence of an attacker $A$ that takes a *clean* test input $\bar{x} \in \mathbb{R}^d$ and returns an *adversarially perturbed* input $A(\bar{x})$. In this work, we focus on attackers that are bounded in the $\ell_\infty$ norm: $A(\bar{x}) \in B_\epsilon(\bar{x})$ for some fixed $\epsilon > 0$. The attacker is successful on a clean input label pair $(\bar{x}, \bar{y})$ if $f(A(\bar{x})) \neq \bar{y}$, or equivalently if $f(A(\bar{x}))_y > f(x^0)_{\bar{y}}$ for some $y \neq \bar{y}$.

We are interested in bounding the error against the *worst-case* attack (we assume the attacker has full knowledge of the neural network). Let $\ell_y^\star(\bar{x}, \bar{y})$ denote the worst-case margin of an incorrect class $y$ that can be achieved in the attack model:

$$\ell_y^\star(\bar{x}, \bar{y}) \overset{\text{def}}{=} \max_{A(x) \in B_\epsilon(\bar{x})} (f(A(x))_y - f(A(x))_{\bar{y}}). \tag{1}$$

A network is *certifiably robust* on $(\bar{x},\bar{y})$ if $\ell_y^\star(\bar{x},\bar{y}) < 0$ for all $y \neq \bar{y}$. Computing $\ell_y^\star(\bar{x},\bar{y})$ for a neural network involves solving a non-convex optimization problem, which is intractable in general. In this work, we study convex relaxations to efficiently compute an upper bound $L_y(\bar{x},\bar{y}) \geq \ell_y^\star(\bar{x},\bar{y})$. When $L_y(\bar{x},\bar{y}) < 0$, we have a certificate of robustness of the network on input $(\bar{x},\bar{y})$.

**Optimization objective.** For a fixed class $y$, the worst-case margin $\ell_y^\star(\bar{x},\bar{y})$ of a neural network $f$ with weights $W$ can be expressed as the following optimization problem. The decision variable is the input $A(x)$, which we denote here by $x^0$ for notational convenience. The quantity we are interested in maximizing is $f(x^0)_y - f(x^0)_{\bar{y}} = (c_y - c_{\bar{y}})^\top x^L$, where $x^L$ is the final layer activation. We set up the optimization problem by jointly optimizing over all the activations $x^0, x^1, x^2, ... x^L$, imposing consistency constraints dictated by the neural network, and restricting the input $x^0$ to be within the attack model. Formally,

$$\ell_y^\star(\bar{x},\bar{y}) = \max_{x^0,...,x^L} \ (c_y - c_{\bar{y}})^\top x^L \tag{2}$$

$$\text{subject to } x^i = \text{ReLU}(W^{i-1} x^{i-1}) \ \text{for } i = 1,2,...,L \qquad \text{(Neural network constraints)}$$

$$\|x_j^0 - \bar{x}_j\|_\infty \leq \epsilon \ \text{for } j = 1,2,...,d \qquad \text{(Attack model constraints)}$$

Computing $\ell_y^\star$ is computationally hard in general. In the following sections, we present how to relax this objective to a convex semidefinite program and discuss some properties of this relaxation.

## 3 Semidefinite relaxations

In this section, we present our approach to obtaining a computationally tractable upper bound to the solution of the optimization problem described in (2).

**Key insight.** The source of the non-convexity in (2) is the ReLU constraints. Consider a ReLU constraint of the form $z = \max(x,0)$. The key observation is that this constraint can be expressed equivalently as the following three linear and quadratic constraints between $z$ and $x$: (i) $z(z-x) = 0$, (ii) $z \geq x$, and (iii) $z \geq 0$. Constraint (i) ensures that $z$ is equal to either $x$ or 0 and constraints (ii) and (iii) together then ensure that $z$ is at least as large as both. This reformulation allows us to replace the non-linear ReLU constraints of the optimization problem in 2 with linear and quadratic constraints, turning it into a quadratically constrained quadratic program (QCQP). We first show how this QCQP can be relaxed to a semidefinite program (SDP) for networks with one hidden layer. The relaxation for multiple layers is a straightforward extension and is presented in Section 5.

### 3.1 Relaxation for one hidden layer

Consider a neural network with one hidden layer containing $m$ nodes. Let the input be denoted by $x \in \mathbb{R}^d$. The hidden layer activations are denoted by $z \in \mathbb{R}^m$ and related to the input $x$ as $z = \text{ReLU}(Wx)$ for weights $W \in \mathbb{R}^{m \times d}$.

Suppose that we have lower and upper bounds $l, u \in \mathbb{R}^d$ on the inputs such that $l_j \leq x_j \leq u_j$. For example, in the $\ell_\infty$ attack model we have $l = \bar{x} - \epsilon \mathbf{1}$ and $u = \bar{x} + \epsilon \mathbf{1}$ where $\bar{x}$ is the clean input. For the multi-layer case, we discuss how to obtain these bounds for the intermediate activations in Section 5.2. We are interested in optimizing a linear function of the hidden layer: $f(x) = c^\top z$, where $c \in \mathbb{R}^m$. For instance, while computing the worst case margin of an incorrect label $y$ over true label $\bar{y}$, $c = c_y - c_{\bar{y}}$.

We use the key insight that the ReLU constraints can be written as linear and quadratic constraints, allowing us to embed these constraints into a QCQP. We can also express the input constraint $l_j \leq x_j \leq u_j$ as a quadratic constraint, which will be useful later. In particular, $l_j \leq x_j \leq u_j$ if and only if $(x_j - l_j)(x_j - u_j) \leq 0$, thereby yielding the quadratic constraint $x_j^2 \leq (l_j + u_j)x_j - l_j u_j$. This gives us the final QCQP below:

$$\ell_y^\star(\bar{x},\bar{y}) = f_{\text{QCQP}} = \max_{x,z} \ c^\top z \tag{3}$$

$$\text{s.t. } z \geq 0, \ z \geq Wx, \ z^2 = z \odot (Wx) \qquad \text{(ReLU constraints)}$$

$$x^2 \leq (l+u) \odot x - l \odot u \qquad \text{(Input constraints)}$$

We now relax the non-convex QCQP (3) to a convex SDP. The basic idea is to introduce a new set of variables representing all linear and quadratic monomials in $x$ and $z$; the constraints in (3) can then be written as *linear* functions of these new variables.

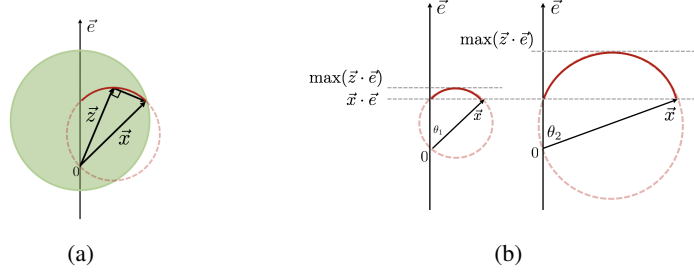

(a)                                              (b)

Figure 1: (a) Plot showing the feasible regions for the vectors $\vec{x}$ (green) and $\vec{z}$ (red). The input constraints restrict $\vec{x}$ to lie within the green circle. The ReLU constraint $\vec{z} \perp \vec{z} - \vec{x}$ forces $\vec{z}$ to lie on the dashed red circle and the constraint $\vec{z} \cdot \vec{e} \geq \vec{x} \cdot \vec{e}$ restricts it to the solid arc. (b) For a fixed value of input $\vec{x} \cdot \vec{e}$, when the angle made by $\vec{x}$ with $\vec{e}$ increases, the arc spanned by $\vec{z}$ has a larger projection on $\vec{e}$ and leading to a looser relaxation. Secondly, for a fixed value of $\vec{x} \cdot \vec{e}$, as $\theta$ increases, the norm $\|\vec{x}\|$ increases and vice versa.

In particular, let $v \stackrel{\text{def}}{=} \begin{bmatrix} 1 \\ x \\ z \end{bmatrix}$. We define a matrix $P \stackrel{\text{def}}{=} vv^\top$ and use symbolic indexing $P[\cdot]$ to index the elements of $P$, i.e $P = \begin{bmatrix} P[1] & P[\mathsf{x}^\top] & P[\mathsf{z}^\top] \\ P[\mathsf{x}] & P[\mathsf{xx}^\top] & P[\mathsf{xz}^\top] \\ P[\mathsf{z}] & P[\mathsf{zx}^\top] & P[\mathsf{zz}^\top] \end{bmatrix}$.

The SDP relaxation of (3) can be written in terms of the matrix $P$ as follows.

$$f_{\text{SDP}} = \max_P \ c^\top P[\mathsf{z}] \tag{4}$$

$$\text{s.t } P[\mathsf{z}] \geq 0, \ P[\mathsf{z}] \geq W P[\mathsf{x}], \ \text{diag}(P[\mathsf{zz}^\top]) = \text{diag}(W P[\mathsf{xz}^\top]) \qquad \text{(ReLU constraints)}$$

$$\text{diag}(P[\mathsf{xx}^\top]) \leq (l+u) \odot P[\mathsf{x}] - l \odot u \qquad \text{(Input constraints)}$$

$$P[1] = 1, \ P \succeq 0 \qquad \text{(Matrix constraints)}.$$

When the matrix $P$ admits a rank-one factorization $vv^\top$, the entries of the matrix $P$ exactly correspond to linear and quadratic monomials in $x$ and $z$. In this case, the ReLU and input constraints of the SDP are identical to the constraints of the QCQP. However, this rank-one constraint on $P$ would make the feasible set non-convex. We instead consider the *relaxed* constraint on $P$ that allows factorizations of the form $P = VV^\top$, where $V$ can be full rank. Equivalently, we consider the set of matrices $P$ such that $P \succeq 0$. This set is convex and is a superset of the original non-convex set. Therefore, the above SDP is a relaxation of the QCQP in 3 with $f_{\text{SDP}} \geq f_{\text{QCQP}}$, providing an upper bound on $\ell_y^\star(\bar{x}, \bar{y})$ that could serve as a certificate of robustness. We note that this SDP relaxation is different from the one proposed in [23], which applies only to neural networks with one hidden layer. In contrast, the construction presented here naturally generalizes to multiple layers, as we show in Section 5. Moreover, we will see in Section 6 that our new relaxation often yields substantially tighter bounds than the approach of [23].

## 4 Analysis of the relaxation

Before extending the SDP relaxation defined in (4) to multiple layers, we will provide some geometric intuition for the SDP relaxation.

### 4.1 Geometric interpretation

First consider the simple case where $m = d = 1$ and $W = c = 1$, so that the problem is to maximize $z$ subject to $z = \text{ReLU}(x)$ and $l \leq x \leq u$. In this case, the SDP relaxation of (4) is as follows:

$$f_{\text{SDP}} = \max_P \ P[\mathsf{z}] \tag{5}$$

$$\text{s.t } P[\mathsf{z}] \geq 0, \ P[\mathsf{z}] \geq P[\mathsf{x}], \ P[\mathsf{z}^2] = P[\mathsf{xz}] \qquad \text{(ReLU constraints)}$$

$$P[\mathsf{x}^2] \leq (l+u) P[\mathsf{x}] - lu \qquad \text{(Input constraints)}$$

$$P[1] = 1, \ P \succeq 0 \qquad \text{(Matrix constraints)}.$$

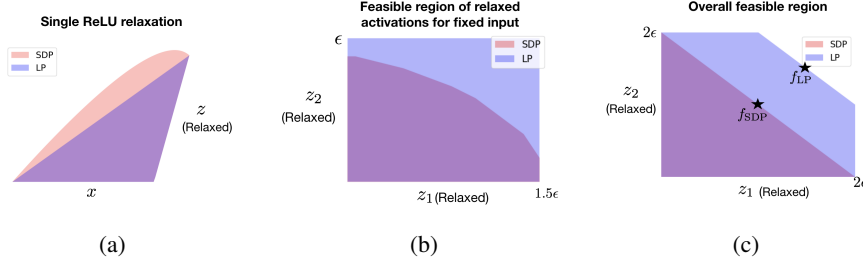

Figure 2: (a) Visualization of the LP and SDP for a single ReLU unit with input $x$ and output $z$. The LP is bounded by the line joining the extreme points. (b) Let $z_1 = \text{ReLU}(x_1 + x_2)$ and $z_2 = \text{ReLU}(x_1 - x_2)$. On fixing the inputs $x_1$ and $x_2$ (both equal to $0.5\epsilon$), we plot the feasible activations of the LP and SDP relaxation. The LP feasible set is a simple product over the independent sets, while the SDP enforces joint constraints to obtain a more complex convex set. (c) We plot the set $(z_1, z_2)$ across all feasible inputs $(x_1, x_2)$ for the same setup as (b) and the objective of maximizing $z_1 + z_2$. We see that $f_{\text{SDP}} < f_{\text{LP}}$.

The SDP operates on a PSD matrix $P$ and imposes linear constraints on the entries of the matrix. Since feasible $P$ can be written as $VV^\top$, the entries of $P$ can be thought of as dot products between vectors, and constraints as operating on these dot products. For the simple example above, $V \overset{\text{def}}{=} \begin{bmatrix} \leftarrow \vec{e} \rightarrow \\ \leftarrow \vec{x} \rightarrow \\ \leftarrow \vec{z} \rightarrow \end{bmatrix}$ for some vectors $\vec{e}, \vec{x}, \vec{z} \in \mathbb{R}^3$. The constraint $P[1] = 1$, for example, imposes $\vec{e} \cdot \vec{e} = 1$ i.e., $\vec{e}$ is a unit vector. The linear monomials $P[\mathsf{x}], P[\mathsf{z}]$ correspond to projections on this unit vector, $\vec{x} \cdot \vec{e}$ and $\vec{z} \cdot \vec{e}$. Finally, the quadratic monomials $P[\mathsf{xz}], P[\mathsf{x}^2]$ and $P[\mathsf{z}^2]$ correspond to $\vec{x} \cdot \vec{z}, \|\vec{x}\|^2$ and $\|\vec{z}\|^2$ respectively. We now reason about the input and ReLU constraints and visualize the geometry (see Figure 1a).

**Input constraints.** The input constraint $P[\mathsf{x}^2] \leq (l + u)P[\mathsf{x}] - lu$ equivalently imposes $\|\vec{x}\|^2 \leq (l + u)(\vec{x} \cdot \vec{e}) - lu$. Geometrically, this constrains vector $\vec{x}$ on a sphere with center at $\frac{1}{2}(l + u)\vec{e}$ and radius $\frac{1}{2}(l - u)$. Notice that this implicitly bounds the norm of $\vec{x}$. This is illustrated in Figure 1a where the green circle represents the space of feasible vectors $\vec{x}$, projected onto the plane containing $\vec{e}$ and $\vec{x}$.

**ReLU constraints.** The constraint on the quadratic terms $(P[\mathsf{z}^2] = P[\mathsf{zx}])$ is the core of the SDP. It says that the vector $\vec{z}$ is perpendicular to $\vec{z} - \vec{x}$. We can visualize $\vec{z}$ on the plane containing $\vec{x}$ and $\vec{e}$ in Figure 1a; the component of $\vec{z}$ perpendicular to this plane is not relevant to the SDP, because it's neither constrained nor appears in the objective. The feasible $\vec{z}$ trace out a circle with $\frac{1}{2}\vec{x}$ as the center (because the angle inscribed in a semicircle is a right angle). The linear constraints restrict $\vec{z}$ to the arc that has a larger projection on $\vec{e}$ than $\vec{x}$, and is positive.

**Remarks.** This geometric picture allows us to make the following important observation about the objective value $\max(\vec{z} \cdot \vec{e})$ of the SDP relaxation. The largest value that $\vec{z} \cdot \vec{e}$ can take depends on the angle $\theta$ that $\vec{x}$ makes with $\vec{e}$. In particular, as $\theta$ decreases, the relaxation becomes tighter and as the vector deviates from $\vec{e}$, the relaxation gets looser. Figure 1b provides an illustration. For large $\theta$, the radius of the circle that $\vec{z}$ traces increases, allowing $\vec{z} \cdot \vec{e}$ to take large values.

That leads to the natural question: For a fixed input value $\vec{x} \cdot \vec{e}$ (corresponding to $x$), what controls $\theta$? Since $\vec{x} \cdot \vec{e} = \|\vec{x}\|\cos\theta$, as the norm of $\vec{x}$ increases, $\theta$ increases. Hence a constraint that forces $\|\vec{x}\|$ to be close to $\vec{x} \cdot \vec{e}$ will cause the output $\vec{z} \cdot \vec{e}$ to take smaller values. Porting this intuition into the matrix interpretation, this suggests that constraints forcing $P[\mathsf{x}^2] = \|\vec{x}\|^2$ to be small lead to tighter relaxations.

## 4.2 Comparison with linear programming relaxation

In contrast to the SDP, another approach is to relax the objective and constraints in (2) to a linear program (LP) [18, 10, 9]. As we will see below, a crucial difference from the LP is that our SDP can "reason jointly" about different activations of the network in a stronger way than the LP can. We briefly review the LP approach and then elaborate on this difference.

**Review of the LP relaxation.** We present the LP relaxation for a neural network with one hidden layer, where the hidden layer activations $z \in \mathbb{R}^m$ are related to the input $x \in \mathbb{R}^d$ as $z = \text{ReLU}(Wx)$. As before, we have bounds $l, u \in \mathbb{R}^d$ such that $l \leq x \leq u$.

In the LP relaxation, we replace the ReLU constraints at hidden node $j$ with a convex outer envelope as illustrated in Figure 2a. The envelope is lower bounded by the linear constraints $z \geq Wx$ and $z \geq 0$. In order to construct the upper bounding linear constraints, we compute the extreme points $s = \min_{l \leq x \leq u} Wx$ and $t = \max_{l \leq x \leq u} Wx$ and construct lines that connect $(s, \text{ReLU}(s))$ and $(t, \text{ReLU}(t))$. The final LP for the neural network is then written by constructing the convex envelopes for each ReLU unit and optimizing over this set as follows:

$$f_{\text{LP}} = \max \ c^\top z \tag{6}$$
$$\text{s.t } z \geq 0, \ z \geq Wx, \qquad\qquad\qquad \text{(Lower bound lines)}$$
$$z \leq \left( \frac{\text{ReLU}(t) - \text{ReLU}(s)}{t - s} \right) \cdot (Wx - s) + \text{ReLU}(s), \qquad \text{(Upper bound lines)}$$
$$l \leq x \leq u \qquad\qquad\qquad\qquad\qquad \text{(Input constraints)}.$$

The extreme points $s$ and $t$ are the optima of a linear transformation (by $W$) over a box in $\mathbb{R}^d$ and can be computed using interval arithmetic. In the $\ell_\infty$ attack model where $l = \bar{x} - \epsilon \mathbf{1}$ and $u = \bar{x} + \epsilon \mathbf{1}$, we have $s_j = W\bar{x} - \epsilon \|W_j\|_1$ and $t_j = W\bar{x} + \epsilon \|W_j\|_1$ for $j = 1, 2, ... m$.

From Figure 2a, we see that for a single ReLU unit taken in isolation, the LP is tighter than the SDP. However, when we have multiple units, the SDP is tighter than the LP. We illustrate this with a simple example in 2 dimensions with 2 hidden nodes (See Figure 2b).

**Simple example to compare the LP and SDP.** Consider a two dimensional example with input $x = [x_1, x_2]$ and lower and upper bounds $l = [-\epsilon, -\epsilon]$ and $u = [\epsilon, \epsilon]$, respectively. The hidden layer activations $z_1$ and $z_2$ are related to the input as $z_1 = \text{ReLU}(x_1 + x_2)$ and $z_2 = \text{ReLU}(x_1 - x_2)$. The objective is to maximize $z_1 + z_2$.

The LP constrains $z_1$ and $z_2$ *independently*. To see this, let us set the input $x$ to a fixed value and look at the feasible values of $z_1$ and $z_2$. In the LP, the convex outer envelope that bounds $z_1$ only depends on the input $x$ and the bounds $l$ and $u$ and is independent of the value of $z_2$. Similarly, the outer envelope of $z_2$ does not depend on the value of $z_1$, and the feasible set for $(z_1, z_2)$ is simply the product of the individual feasible sets.

In contrast, the SDP has constraints that couple $z_1$ and $z_2$. As a result, the feasible set of $(z_1, z_2)$ is *a strict subset* of the product of the individual feasible sets. Figure 2b plots the LP and SDP feasible sets $[z_1, z_2]$ for $x = [\frac{\epsilon}{2}, \frac{\epsilon}{2}]$. Recall from the geometric observations (Section 4.1) that the arc of $\vec{z_1}$ depends on the configuration of $\vec{x_1} + \vec{x_2}$, while that of $\vec{z_2}$ depends on $\vec{x_1} - \vec{x_2}$. Since the vectors $\vec{x_1} + \vec{x_2}$ and $\vec{x_1} - \vec{x_2}$ are dependent, the feasible sets of $\vec{z_1}$ and $\vec{z_2}$ are also dependent on each other. An alternative way to see this is from the matrix constraint that $P \succeq 0$ in 4. This matrix constraint does *not* factor into terms that decouple the entries $P[\mathsf{z_1}]$ and $P[\mathsf{z_2}]$, hence $z_1$ and $z_2$ cannot vary independently.

When we reason about the relaxation over all feasible points $x$, the joint reasoning of the SDP allows it to achieve a better objective value. Figure 2c plots the feasible sets $[z_1, z_2]$ over all valid $x$ where the optimal value of the SDP, $f_{\text{SDP}}$, is less than that of the LP, $f_{\text{LP}}$.

We can extend the preceding example to exhibit a dimension-dependent gap between the LP and the SDP for random weight matrices. In particular, for a random network with $m$ hidden nodes and input dimension $d$, with high probability, $f_{\text{LP}} = \Theta(md)$ while $f_{\text{SDP}} = \Theta(m\sqrt{d} + d\sqrt{m})$. More formally:

**Proposition 1.** *Suppose that the weight matrix $W \in \mathbf{R}^{m \times d}$ is generated randomly by sampling each element $W_{ij}$ uniformly and independently from $\{-1, +1\}$. Also let the output vector $c$ be the all-1s vector, $\mathbf{1}$. Take $\bar{x} = 0$ and $\epsilon = 1$. Then, for some universal constant $\gamma$,*

$$f_{\text{LP}} \geq \frac{1}{2} md \ \text{almost surely, while}$$

$$f_{\text{SDP}} \leq \gamma \cdot (m\sqrt{d} + d\sqrt{m}) \ \text{with probability } 1 - \exp(-(d+m)).$$

We defer the proof of this to Section A.

## 5 Multi-layer networks

The SDP relaxation to evaluate robustness for multi-layer networks is a straightforward generalization of the relaxation presented for one hidden layer in Section 3.1.

|  | Grad-NN [23] | LP-NN [29] | PGD-NN |
|---|---|---|---|
| PGD-attack | 15% | 18% | 9% |
| SDP-cert (this work) | **20%** | **20%** | **18%** |
| LP-cert | 97% | 22% | 100% |
| Grad-cert | 35% | 93% | n/a |

Table 1: Fraction of non-certified examples on MNIST. Different certification techniques (rows) on different networks (columns). SDP-cert is consistently better than other certificates. All numbers are reported for $\ell_\infty$ attacks at $\epsilon = 0.1$.

## 5.1 General SDP

The interactions between $x^{i-1}$ and $x^i$ in (2) (via the ReLU constraint) are analogous to the interaction between the input and hidden layer for the one layer case. Suppose we have bounds $l^{i-1}, u^{i-1} \in \mathbb{R}^{m_{i-1}}$ on the inputs to the ReLU units at layer $i$ such that $l^{i-1} \leq x^{i-1} \leq u^{i-1}$. We discuss how to obtain these bounds and their significance in Section 5.2. Writing the constraints for each layer iteratively gives us the following SDP:

$$f_y^{\text{SDP}}(\bar{x}, \bar{y}) = \max_P \ (c_y - c_{\bar{y}})^\top P[\mathsf{x}^\mathsf{L}] \tag{7}$$

s.t. for $i = 1, ..., L$

$$P[\mathsf{x}^\mathsf{i}] \geq 0, \ P[\mathsf{x}^\mathsf{i}] \geq W^{i-1} P[\mathsf{x}^{\mathsf{i}-1}],$$

$$\text{diag}(P[\mathsf{x}^\mathsf{i}(\mathsf{x}^\mathsf{i})^\top]) = \text{diag}(W P[\mathsf{x}^{\mathsf{i}-1}(\mathsf{x}^\mathsf{i})^\top]), \qquad \text{(ReLU constraints for layer } i)$$

$$\text{diag}(P[\mathsf{x}^{\mathsf{i}-1}(\mathsf{x}^{\mathsf{i}-1})^\top]) \leq (l^{i-1} + u^{i-1}) \odot P[\mathsf{x}^{\mathsf{i}-1}] - l^{i-1} \odot u^{i-1}, \quad \text{(Input constraints for layer } i)$$

$$P[\mathbb{1}] = 1, \ P \succeq 0 \qquad \text{(Matrix constraints).}$$

## 5.2 Bounds on intermediate activations

From the geometric interpretation of Section 4.1, we made the important observation that adding constraints that keep $P[\mathsf{x}^2]$ small aid in obtaining tighter relaxations. For the multi-layer case, since the activations at layer $i-1$ act as input to the next layer $i$, adding constraints that restrict $P[(\mathsf{x}_j^\mathsf{i})^2]$ will lead to a tighter relaxation for the overall objective. The SDP automatically obtains some bound on $P[(\mathsf{x}_j^\mathsf{i})^2]$ from the bounds on the input, hence the SDP solution is well-defined and finite even without these bounds. However, we can tighten the bound on $P[(\mathsf{x}_j^\mathsf{i})^2]$ by relating it to the linear monomial $P[(\mathsf{x}_j^\mathsf{i})]$ via bounds on the value of the activation $x_j^i$. One simple way to obtain bounds on activations $x_j^i$ is to treat each hidden unit separately, using simple interval arithmetic to obtain

$$l^0 = \bar{x} - \epsilon \mathbb{1} \ \text{(Attack model)}, \qquad u^0 = \bar{x} + \epsilon \mathbb{1} \ \text{(Attack model)}, \tag{8}$$

$$l^i = [W^{i-1}]_+ l^{i-1} + [W^{i-1}]_- u^{i-1}, \qquad u^i = [W^{i-1}]_+ u^{i-1} + [W^{i-1}]_- l^{i-1},$$

where $([M]_+)_{ij} = \max(M_{ij}, 0)$ and $([M]_-)_{ij} = \min(M_{ij}, 0)$.

In our experiments on real networks (Section 6), we observe that these simple bounds are sufficient to obtain good certificates. However tighter bounds could potentially lead to tighter certificates.

## 6 Experiments

In this section, we evaluate the performance of our certificate (7) on neural networks trained using different robust training procedures, and compare against other certificates in the literature.

**Networks.** We consider feedforward networks that are trained on the MNIST dataset of handwritten digits using three different robust training procedures.

**1. Grad-NN.** We use the two-layer network with 500 hidden nodes from [23], obtained by using an SDP-based bound on the gradient of the network (different from the SDP presented here) as a regularizer. We obtained the weights of this network from the authors of [23].

**2. LP-NN.** We use a two-layer network with 500 hidden nodes (matching that of Grad-NN) trained via the LP-based robust training procedure of [29]. The authors of [29] provided the weights.

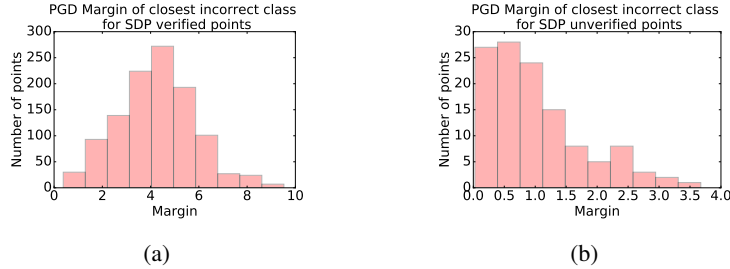

Figure 3: Histogram of PGD margins for (a) points that are certified by the SDP and (b) points that are not certified by the SDP.

**3. PGD-NN.** We consider a fully-connected network with four layers containing 200,100 and 50 hidden nodes (i.e., the architecture is 784-200-100-50-10). We train this network using adversarial training [12] against the strong PGD attack [21]. We train to minimize a weighted combination of the regular cross entropy loss and adversarial loss. We tuned the hyperparameters based on the performance of the PGD attack on a holdout set. The stepsize of the PGD attack was set to $0.1$, number of iterations to $40$, perturbation size $\epsilon = 0.3$ and weight on adversarial loss to $\frac{1}{3}$.

The training procedures for SDP-NN and LP-NN yield certificates of robustness (described in their corresponding papers), but the training procedure of PGD-NN does not. Note that all the networks are "foreign networks" to our SDP, as their training procedures do not incorporate the SDP relaxation.

**Certification procedures.** Recall from Section 2 that an upper bound on the maximum incorrect margin can be used to obtain certificates. We consider certificates from three different upper bounds.

**1. SDP-cert.** This is the certificate we propose in this work. This uses the SDP upper bound that we defined in Section 5. The exact optimization problem is presented in (7) and the bounds on intermediate activations are obtained using the interval arithmetic procedure presented in (8).

**2. LP-cert.** This uses the upper bound based on the LP relaxation discussed in Section 4.2 which forms the basis for several existing works on scalable certification [9, 10, 28, 29]. The LP uses layer-wise bounds for intermediate nodes, similar to $l_i, u_i$ in our SDP formulation (7). For Grad-NN and LP-NN with a single hidden layer, the layerwise bounds can be computed *exactly* using interval arithmetic. For the four-layer PGD-NN, in order to have a fair comparison with SDP-cert, we use the same procedure (interval arithmetic) (8).

**3. Grad-cert.** We use the upper bound proposed in [23]. This upper bound is based on the maximum norm of the gradient of the network predictions and only holds for two-layer networks.

Table 1 presents the performance of the three different certification procedures on the three networks. For each certification method and network, we evaluate the associated upper bounds on the same 1000 random test points and report the fraction of points that were not certified. Computing the exact worst-case adversarial error is not computationally tractable. Therefore, to provide a comparison, we also compute a lower bound on the adversarial error—the error obtained by the PGD attack.

**Performance of proposed SDP-cert.** SDP-cert provides non-vacuous certificates for *all* networks considered. In particular, we can certify that the four layer PGD-NN has an error of at most $18\%$ at $\epsilon = 0.1$. To compare, a lower bound on the robust error (PGD attack error) is $9\%$. On the two-layer networks, SDP-cert improves the previously-known bounds. For example, it certifies that Grad-NN has an error of at most $20\%$ compared to the previously known $35\%$. Similarly, SDP-cert improves the bound for LP-NN from $22\%$ to $20\%$.

The gap between the lower bound (PGD) and upper bound (SDP) is because of points that cannot be misclassified by PGD but are also not certified by the SDP. In order to further investigate these points, we look at the margins obtained by the PGD attack to estimate the robustness of different points. Formally, let $x_{\text{PGD}}$ be the adversarial example generated by the PGD attack on clean input $\bar{x}$ with true label $\bar{y}$. We compute $\min_{y \neq \bar{y}}[f(x_{\text{PGD}})_{\bar{y}} - f(x_{\text{PGD}})_y]$, the margin of the closest incorrect class. A small value indicates that the $x_{\text{PGD}}$ was close to being misclassified. Figure 3 shows the histograms of the above PGD margin. The examples which are not certified by the SDP have much smaller margins than those examples that are certified: the average PGD margin is 1.2 on points that are not certified

and 4.5 on points that are certified. From Figure 3, we see that a large number of the SDP uncertified points have very small margin, suggesting that these points might be misclassified by stronger attacks.

**Remark.** As discussed in Section 5, we could consider a version of the SDP that does not include the constraints relating linear and quadratic terms at the intermediate layers of the network. Empirically, such an SDP produces vacuous certificates ($>90\%$ error). Therefore, these constraints at intermediate layers play a significant role in improving the empirical performance of the SDP relaxation.

**Comparison with other certification approaches.** From Table 1, we observe that SDP-cert consistently performs better than both LP-cert and Grad-cert for all three networks.

Grad-cert and LP-cert provide vacuous ($>90\%$ error) certificates on networks that are not trained to minimize these certificates. This is because these certificates are tight only under some special cases that can be enforced by training. For example, LP-cert is tight when the ReLU units do not switch linear regions [29]. While a typical input causes only $20\%$ of the hidden units of LP-NN to switch regions, $75\%$ of the hidden units of Grad-NN switch on a typical input. Grad-cert bounds the gradient uniformly across the *entire* input space. This makes the bound loose on arbitrary networks that could have a small gradient only on the data distribution of interest.

**Comparison to concurrent work [26].** A variety of robust MNIST networks are certified by Tjeng and Tedrake [26]. On Grad-NN, their certified error is $30\%$ which is looser than our SDP certified error ($20\%$). They also consider the CNN counterparts of LP-NN and PGD-NN, trained using the procedures of [29] and [21]. The certified errors are $4.4\%$ and $7.2\%$ respectively. This reduction in the errors is due to the CNN architecture. Further discussion on applying our SDP to CNNs appears in Section 7.

**Optimization setup.** We use the YALMIP toolbox [19] with MOSEK as a backend to solve the different convex programs that arise in these certification procedures. On a 4-core CPU, the average SDP computation took around 25 minutes and the LP around 5 minutes per example.

# 7    Discussion

In this work, we focused on fully connected feedforward networks for computational efficiency. In principle, our proposed SDP can be directly used to certify convolutional neural networks (CNNs); unrolling the convolution would result in a (large) feedforward network. Naively, current off-the-shelf solvers cannot handle the SDP formulation of such large networks. Robust training on CNNs leads to better error rates: for example, adversarial training against the PGD adversary on a four-layer feedforward network has error $9\%$ against the PGD attack, while a four-layer CNN trained using a similar procedure has error less than $3\%$ [21]. An immediate open question is whether the network in [21], which has so far withstood many different attacks, is truly robust on MNIST. We are hopeful that we can scale up our SDP to answer this question, perhaps borrowing ideas from work on highly scalable SDPs [1] and explicitly exploiting the sparsity and structure induced by the CNN architecture.

Current work on certification of neural networks against adversarial examples has focused on perturbations bounded in some norm ball. In our work, we focused on the common $\ell_\infty$ attack because the problem of securing multi-layer ReLU networks remains unsolved even in this well-studied attack model. Different attack models lead to different constraints only at the input layer; our SDP framework can be applied to any attack model where these input constraints can be written as linear and quadratic constraints. In particular, it can also be used to certify robustness against attacks bounded in $\ell_2$ norm. [13] provide alternative bounds for $\ell_2$ norm attacks based on the local gradient.

Guarantees for the bounded norm attack model in general are sufficient but not necessary for robustness against adversaries in the real world. Many successful attacks involve inconspicuous but clearly visible perturbations [11, 24, 6, 4], or large but semantics-preserving perturbations in the case of natural language [15]. These perturbations do not currently have well-defined mathematical models and present yet another layer of challenge. However, we believe that the mathematical ideas we develop for the bounded norm will be useful building blocks in the broader adversarial game.

**Reproducibility.** All code, data and experiments for this paper are available on the Codalab platform at `https://worksheets.codalab.org/worksheets/0x6933b8cdbbfd424584062cdf40865f30/`.

**Acknowledgements.** This work was partially supported by a Future of Life Institute Research Award and Open Philanthrophy Project Award. JS was supported by a Fannie & John Hertz Foundation Fellowship and an NSF Graduate Research Fellowship. We thank Eric Wong for providing relevant experimental results. We are also grateful to Moses Charikar, Zico Kolter and Eric Wong for several helpful discussions and anonymous reviewers for useful feedback.

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
