[Supplementary Material]

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

# A Proof of Proposition 1

We first lower bound the LP value $f_{\mathrm{LP}}$, and then upper bound the SDP value $f_{\mathrm{SDP}}$.

**Part 1: Lower-bounding $f_{\mathrm{LP}}$.** It suffices to exhibit a feasible solution for the constraints. Note that for a given hidden unit $i$, we have $s_i = -\|W_i\|_1$ and $t_i = \|W_i\|_1$. In particular, at $x = 0$ a feasible value for $z_i$ is $\frac{1}{2}\|W_i\|_2$.

For this feasible value of $(x, z)$, we get that $c^\top z = \sum_{i=1}^m \frac{1}{2}\|W_i\|_1 = \frac{1}{2}\sum_{i,j}|W_{ij}|$. In other words, $f_{\mathrm{LP}}$ is at least half the element-wise $\ell_1$-norm of $W$. Since $W$ is a random sign matrix we have $|W_{ij}| = 1$ for all $i, j$, hence $f_{\mathrm{LP}} \geq \frac{1}{2}md$ with probability 1.

**Part 2: Upper-bounding $f_{\mathrm{SDP}}$.** We start by exhibiting a general upper bound on $f_{\mathrm{SDP}}$ implied by the constraints:

**Lemma 1.** *For any weight matrices $W$ and $c$, we have $f_{SDP} \leq \sqrt{d}\|W\|_2\|c\|_2$, where $\|W\|_2$ is the operator norm of $W$.*

The proof of Lemma 1 is given later in this section. To apply the lemma, note that in our case $\|c\|_2 = \sqrt{m}$, while $\|W\|_2 \leq \gamma \cdot (\sqrt{m} + \sqrt{d} + \sqrt{\log(1/\delta)})$ with probability $1 - \delta$, for some universal constant $\gamma$ (see Theorem 5.39 of [27]). Therefore, Lemma 1 yields the bound $f_{\mathrm{SDP}} \leq \gamma \cdot \sqrt{md} \cdot (\sqrt{m} + \sqrt{d} + \sqrt{m+d}) \leq 2\gamma \cdot (m\sqrt{d} + d\sqrt{m})$ with probability $1 - \exp(-(m+d))$, as claimed.

## A.1 Proof of Lemma 1

First note that since $\begin{bmatrix} P[1] & P[\mathsf{z}^\top] \\ P[\mathsf{z}] & P[\mathsf{z}\mathsf{z}^\top] \end{bmatrix} \succeq 0$, we have $P[\mathsf{z}]P[\mathsf{z}]^\top \preceq P[\mathsf{z}\mathsf{z}^\top]$ by Schur complements, and in particular $\|P[\mathsf{z}]\|_2^2 \leq \mathrm{tr}P[\mathsf{z}\mathsf{z}^\top]$ (by taking the trace of both sides).

Using this, and letting $\|\cdot\|_*$ denote the nuclear norm (sum of singular values), we have

$$c^\top P[\mathsf{z}] \leq \|c\|_2\|P[\mathsf{z}]\|_2 \tag{9}$$

$$\leq \|c\|_2\sqrt{\mathrm{tr}P[\mathsf{z}\mathsf{z}^\top]}. \tag{10}$$

But we also have

$$\mathrm{tr}P[\mathsf{z}\mathsf{z}^\top] = \sum_{i=1}^m P[\mathsf{z}_\mathsf{i}^2] \tag{11}$$

$$= \sum_{i=1}^m W_i^\top P[\mathsf{x}\mathsf{z}_\mathsf{i}] \tag{12}$$

$$= \mathrm{tr}(WP[\mathsf{x}\mathsf{z}^\top]) \tag{13}$$

$$\overset{(i)}{\leq} \|W\|_2\|P[\mathsf{x}\mathsf{z}^\top]\|_* \tag{14}$$

$$\overset{(ii)}{\leq} \|W\|_2\sqrt{\mathrm{tr}(P[\mathsf{x}\mathsf{x}^\top])\mathrm{tr}(P[\mathsf{z}\mathsf{z}^\top])} \tag{15}$$

$$\overset{(iii)}{\leq} \|W\|_2\sqrt{d}\sqrt{\mathrm{tr}P[\mathsf{x}\mathsf{x}^\top]}. \tag{16}$$

Here (i) is Hölder's inequality, and (iii) uses the fact that $P[\mathsf{x}_\mathsf{j}^2] \leq 1$ for all $j$ (due to the constraints imposed by $l$ and $u$).

Solving for $\mathrm{tr}P[\mathsf{z}\mathsf{z}^\top]$, we obtain the bound $\mathrm{tr}P[\mathsf{z}\mathsf{z}^\top] \leq \|W\|_2^2 d$. Plugging back into the preceding inequality, we obtain $c^\top P[\mathsf{z}] \leq \|c\|_2\|W\|_2\sqrt{d}$, as was to be shown.