[Reviews · NeurIPS 2018]

Reviewer 1



This paper studies certified defenses against adversarial attacks on neural networks and proposes a Semidefinite Programming (SDP) relaxation for multi-layer networks with Rectified Linear Unit (Relu) activations and adversaries bounded by infinity norm ball. The paper builds on the linear programming approach of Kolter and Wong, where the authors express the relu constraint as an equivalent quadratic constraint. Writing the quadratic constraint as a SDP is then straight forward. The certification upper-bound is then obtained by considering a Lagrangian relaxation of this constrained problem. My main issue with the paper is that while the SDP relaxation introduced may be novel, it is not clear that this will ever scale to anything beyond simple toy problems and toy architectures. Besides this, the SDP relaxation crucially relies on relu non-linearities and it is not clear how they can extend this to other non-linearities such as sigmoid, tanh etc or with other non-linearities typically used by practitioners such as batch normalization etc. The other issue is the usefulness of studying l_infinity bounded adversaries: no real world adversary is going to restrict themselves to a fixed epsilon ball, and even if they do, what choice of epsilon should one choose for the certificate? In light of this, I think this paper is more suitable for a theory conference such as COLT and not very well suited for NIPS.

Reviewer 2



This paper proposes a new method for certifying adversarial robustness of ReLU networks. Here "certified robustness" is achieved when the "worst-case margin" in Eq (2) is negative. Computing this "worst-case margin" is intractable and the authors instead upper bound this worst-case margin by formulating a quadratically constrained quadratic program, and then relaxing it to a convex semidefinite program. Experimental results, on small fully connected neural networks, show that this SDP-cert bound can provide tighter bounds on networks trained differently. The explanation of the general idea of SDP-cert is fairly clear, and the authors also explained its relationships to previous work on LP-cert [11] and dual verification [5]. The new results on certifying PGD-NN is significant. However, there are places where the presentation is not very clear and some of them actually affect understanding. In particular, I would like some clarifications for the following questions 1) If I'm understanding it correctly, non-convexity in Eq (4) is from z_j^2=z_j(W_j^T x). How does SDP relaxation make it convex? 2) Figure 2b and 2c, what're the coordinates of the bottom left corners? 3) Dimensionality of P in Eq (8), is it a n-by-n matrix where n=\sum_{i=1}^L m_i ? and how does the computation scales with the size the of the network? 4) around line 212~220 re: layerwise bounds l and u, does values of l and u only affect the optimization speed? or it actually affect the optimal value f_{SDP}? how much difference does it make for not having l and u, having looser l and u and having tighter l and u? 5) just to confirm 25 minutes (line 252) is a single example? and is there a big difference between time needed for PGD-NN, LP-NN and MaxGrad-NN? Also I think the paper should include the following contents to be complete 1) line 278-282, there should be numbers/table/figures accompanied, the current form is not precise and not convincing 2) what is the main bottleneck to verify larger network? computation? memory? what are the computational and memory complexities? 3) what happens to the SDP if epsilon is larger? say 0.3? will the bound become more vacuous? or will be any computation issues? also how does it compare to other methods? There are also many minor issues - line 68: non-negative should be non-positive? - line 129: P=V^T V, not consistent with previous text at line 117 where P=V V^T - line 136: The radius is probably (u_1-l_1)/2? instead of l_1*u_1? - Figure 2: colors are confusing because of the inconsistency between figure and the legend - line 146: “increases” should be “decreases” ? - line 274: tighte - a few failed latex ???, e.g. line 46 - line 214~215, grammar - ref [11] [22] are the same paper - line 206 typo in the inequalities I would recommend acceptance if these issues can be resolved.

Reviewer 3



Summary =============== The paper proposes a method to certify the adversarial robustness of a given ReLU neural network. The main difficulty in certifying these networks is dealing with the non-convex ReLU constraints. The proposed method formulates these as quadratic constraints to obtain a quadratic program that is then relaxed to an SDP. The idea is natural and well-executed. An important feature of this approach is that the verified network does not need to be "verification-aware" (i.e. trained towards the goal of being verifiable by this procedure). The experimental results are compelling, constructing the first certificate of robustness for verification-oblivious MNIST models w.r.t. moderate sized L_infty perturbations. This is a clear step of progress towards robust and verifiable modes, thus I recommend acceptance. Quality =============== The problem studied is important and has received significant attention recently. The method is natural and effective. All necessary experiments are conducted. There is a good amount of analysis explaining how the method differs in terms of approximation tightness from the corresponding LP relaxations qualitatively and quantitatively. Clarity =============== The paper is well-written. The approach and analysis is clear. There are a few places where writing looks sloppy (e.g. beginning of Section 3.2) but this has no effect on the overall readability. Originality =============== The approach is novel and is clearly differentiated from prior work. While SDP formulations for NN verification where also proposed in Raghunathan et al [17] the approach is essentially completely different. In fact, it's unclear if these SDP relaxation could scale reasonably to more than a single layer. Significance =============== Training adversarially robust models is an important topic. There are currently three approaches towards evaluating these models: a) empirical, b) exact verification, c) efficient upper bounds. This paper makes a significant step in direction c). In fact it is the first method to verify independently trained networks on MNIST against moderate-sized adversaries (eps=0.1). Additional comments =============== I don't understand the choice of using a 2-layer neural network for the LP-NN. The paper of Kolter and Wong [11] uses larger models and achieves a provable error rate of at most 8% with LP-cert (half the error rate you report in Table 1). Why wasn't the original model from their paper used? Comments to the authors =============== 66. "..and is likely intractable.": In fact the ReLUplex paper shows that verifying ReLU networks is NP-hard in general. 274. "tighte", typo References: you list Madry et al. twice. ** UPDATE ** ============== While I stand behind my original score based on the technical contributions of the paper, I agree with Reviewer 2 that several improvements to the exposition are in order. I believe that the authors should implement these improvements independent of the final decision. I personally would like to see experimental results for LP-NN applied to large models.